# Effects of External Environments on the Fixed Elongation and Tensile Properties of the VAE Emulsion–Cement Composite Joint Sealant

**DOI:** 10.3390/ma13143233

**Published:** 2020-07-21

**Authors:** Chuanxin Lou, Jinyu Xu, Sinuo Liu, Tengjiao Wang, Weibo Ren

**Affiliations:** 1Aeronautics Engineering College, Air Force Engineering University, Xi’an 710038, China; xujinyu_afeu@163.com (J.X.); wtengjiao83087@163.com (T.W.); Ren159342@163.com (W.R.); 2College of Mechanics and Civil Architecture, Northwest Polytechnic University, Xi’an 710072, China; 3College of Architecture, Xi’an University of Architecture and Technology, Xi’an 710055, China; liu374812@163.com

**Keywords:** VAE emulsion–cement composite material, fixed elongation property, tensile property, temperature, water soaking, ultraviolet radiation, durability

## Abstract

Joint sealant is affected by various environmental factors in service, such as different temperatures, water soaking, ultraviolet and so on. In this paper, the VAE emulsion–cement compositejoint sealant was pretreated under multiple simulation environments. Thereafter, the degradation rules of fixed elongation and tensile properties of joint sealants at different mix proportions were systemically investigated under the action of external environments (temperature, water soaking and ultraviolet), and the influence mechanisms of diverse environmental factors were analyzed. The research results suggested that, under the action of external environments, the VAE emulsion–cement composite joint sealants exhibited degradation effects to varying degrees. After the addition of plasticizer, the joint sealants had reduced cohesion strength in low temperature environment and enhanced flexible deformability. The addition of water repellent improved the water resistance of joint sealants. Meanwhile, adding ultraviolet shield agent partially improved the ultraviolet radiation aging resistance. A greater powder–liquid ratio led to the lower flexibility of joint sealants, but superior water resistance and ultraviolet radiation aging resistance.

## 1. Introduction

Airfield pavement joint sealants are used to connect the airport pavement slabs. Therefore, the properties of sealing joint materials used at seams and the achieved joint sealing effect directly affect the usability of the airfield pavement [1]. At present, the most commonly used pavement joint sealants are polymer materials, such as polyurethane, polysulfide and silicone [2,3,4,5]. Such materials have superior deformation performances and bonding properties and are convenient for construction. However, these polymer joint sealants are affected in service by different environmental factors, resulting in strength reduction and adhesion decline of joint sealants. This caninduce the phenomena of airfield pavementpumping and faulting of slab ends, affecting the normal use of the pavement [6,7]. Consequently, scholars at home and abroad have carried out extensive investigation on the durability of organic joint sealants. Chen et al. [8] and Liu et al. [9] respectively prepared a novel self-leveling silicone rubber joint sealant. They tested the changes in their mechanical properties under different working conditions (water soaking, freeze-thaw cycles, cold-drawn hot-pressing fatigue load) and the chemical stability in multiple corrosive environments. Ding et al. [10] conducted tensile test, hardness test, thermogravimetric analysis and differential thermal analysis to investigate the aging performance of silicone and polyurethane joint sealants under ultraviolet radiation. White et al. [11] studied the durability of joint sealants under the single or combined effect of different environmental factors like temperature and humidity. Li et al. [12] and Liu et al. [13] reviewed the performance characteristics, requirements, research and development situation of existing silicone joint sealants, and proposed the technical routes to enhance the durability of joint sealant. Most of the above studies improve the durability of organic joint sealants through changing the mix proportion and adding admixtures. However, the use of organic polymer joint sealants alone cannotmeet the working requirements of joint sealants in different environments. Therefore, it is necessary to introduce new components to further optimize and modify the organic polymer joint sealants.

Polymer–cement composite is a kind of novel material based on the “organic–inorganic composite” concept, which not only has good bonding propertiesand excellent deformation of organic material, but also possesses the high strength and high durability of inorganic material [14,15,16]. Moreover, the addition of other admixtures allowsfor secondary modification of the polymer–cement composite material, so that it has good bonding deformation property and superb durability [17,18,19]. To date, some scholars have extensively investigated the polymer–cement composite materials. For instance, Ribeiro et al. [20] and Pascalet al. [21] carried out compression test to investigate the mechanical properties of butylbenzene polymer-modified mortar with different polymer-to-cement weight ratios. Almeida et al. [22] studied the bonding strength and microstructure of styrene-acrylic polymer-modified mortar. The results suggested that, after polymer modification, the average pore size inside mortar decreased, the compactness at bonding interface increased, and the bonding strength elevated. Xue et al. [23] used styrene-acrylic polymer and cement as the main base materials to prepare a kind of waterproof with good tensile performance and impermeability. Yang et al. [24] found that, with the increase in the added proportion of VAE re-dispersible polymer powder, both the tensile strength and tensile modulus of polymer-modified cement-based composite joint sealant increased significantly. The above studies focus on the mechanical properties of polymer–cement composite materials, while few studies investigate theeffects of external environments. However, in practical environment, joint sealants are affected by multiple environmental factors, such as temperatures and water soaking. Therefore, it is significant to explore the mechanical properties of polymer–cementcomposite joint sealants in different environments. 

This study investigated the fixed elongation and tensile properties of the VAE emulsion–cement composite joint sealant under different environments (temperature, water soaking and ultraviolet radiation). In addition, the influence rules in different environments were summarized and the mechanisms of action of various environmental factorswere intensively analyzed. The field test was conducted to analyze the durability of the VAE emulsion–cement composite joint sealant. The research results are significant to further optimize the mix proportion of the VAE emulsion–cement composite joint sealant and improve the working performance of joint sealants in different environments.

## 2. Tests

### 2.1. Materials

The raw materials used to prepare the VAE emulsion–cement composite joint sealants included: VAE emulsion (Celanese company, Dallas, TX, USA; technical indexes are shown in Table 1); the cement used was 42.5R ordinary Portland cement(LantianYaobai factory, Xi’an, China); talc powder (Liaoning Wantong Powder Co., Ltd., Yingkou, China; 600 mesh, 60% SiO_2_ and 30% MgO); quartz powders (Shanxi Jiamei Mining Co., Ltd., Hanzhong, China; 300 mesh, >99% SiO_2_); SN-Defoamer 345 off white silicone defoamer (Sannopco company, Tokyo, Japan; technical indexes are displayed in Table 2); SN-Dispersant 5040 dispersing agent (Sannopco company, Tokyo, Japan; technical indexes are displayed in Table 3); DN-12film-forming additive (Tianyin Chemical Co., Ltd., Yixing, China; active ingredient >99%); KH-550 silicone coupling agent (Yingchu Chemical company, Jinan, China; technical indexes are displayed in Table 4); the plasticizer was the dioctyl phthalate (Tianjin ZhiyuanChemical Reagent Co., Ltd., Tianjin, China; purity ≥99%); Elotex SEAL81 silicone water repellent (Shanghai Akzonobel Special Chemistry Co., Ltd., Shanghai, China; technical indexes are displayed in Table 5); ferric oxide (Xinzheng Chemical Co., Ltd., Weifang, China) was used as the ultraviolet shield agent; and all water used in the experiment was clean tap water in the laboratory. 

### 2.2. Design of Mix Proportions

The working performance of joint sealant under different working conditions is mainly related to the organic compositional change inside the joint sealant. Among them, the mix proportion K was the basic mix proportion, while mix proportion Y was the control group to examine the influence of VAE emulsion compositional content changes on the joint sealant durability. Plasticizer is good for improving the low temperature flexibility of polymer–cement composite material. As a result, mix proportion Z examined the effect of plasticizer addition on the low temperature performance of joint sealant. Organic silicone powders have superb hydrophobicity. Thus, mix proportion S was designed to investigate the influence of water repellent addition on the water resistance of joint sealant. Moreover, ferric oxide powders have good ultraviolet shield effect. Therefore, mix proportion M was designed to investigate the influence of ultraviolet shield agent addition on the ultraviolet radiation aging resistance of joint sealant. The mix proportionsof the VAE emulsion–cement composite materials are presented in Table 6.

### 2.3. Preparation of Test Specimens

The procedure for preparing the VAE emulsion–cement composite joint sealant test specimen was shown below.

(1)The VAE emulsion and dispersing agent were poured into the blender simultaneously to stir for 2 min, so that the VAE emulsion dispersed evenly. Subsequently, the defoamer, coalescing agent, (water repellent, plasticizer) and coupling agent were added to stir for 5 min to obtain the even mixed solution.(2)Then, cement, quartz powders, (ultraviolet shield agent) and talc powders were mixed together and stirred evenly, then the mixture was slowly added into the prepared mixed solution to stir at low velocity for 5 min and then at high velocity for 10 min. Thus, the evenly mixed VAE emulsion–cement composite joint sealant with stable properties was obtained.(3)Later, the prepared VAE emulsion–cement composite joint sealant was poured into the mould (as shown in Figure 1), followed by curing for 28 days under standard conditions (temperature, 21–25 °C; relative humidity, 45–55%). Eventually, the joint sealant test specimens for tests were obtained after removal from the mold. Both sides of test specimens were prefabricated cement mortar blocks, as presented in Figure 2.

### 2.4. Test Methods

Joint sealants under normal temperature (20 ± 2 °C, with no other pretreatment) were used as the control group. Specifically, the effects of environmental factors on the mechanical properties of joint sealants were considered. Among them, temperatures were set at the low temperatures of −20 °C and −10 °C; water soaking included short-term, long-term water soaking, and dry–wet cycle treatments; ultraviolet radiation aging included short-term and long-term ultraviolet radiation treatments. Then the specimens after pretreatment are used for fixed elongation test and tensile test respectively. The tests methods were carried out according to the Chinese standard [25] (GB/T 13477-2018, Test Methods for Building Sealants). Eight specimens were allocated to eachmix proportion under each condition, three for fixed elongation test, three for tensile test and the remaining two specimens were used as spare parts. The average of three repeated tests was taken as the final test result of each group. The pretreatment methods for test specimens under different environments were shown below.

(1)Low temperature treatments: the low temperature treatments of test specimens were performed in the BPHJS-060B high–low temperature cycle chamber (Shanghai Yiheng Science and Technology Ltd., Shanghai, China), as shown in Figure 3. In the tests, the test specimens were placed in the −20 °C and −10 °C test chambers at constant temperature for 24 h, respectively. Thereafter, the test specimens were taken out to immediately carry out fixed elongation and tensile tests. For specimens used in fixed elongation test, their fixed elongation maintenance process and elastic recovery process after pad removal were conducted at the original pretreatment temperature.(2)Water soaking treatments: in the tests, the test specimens were immersed into the container filled with tap water at normal temperature. They were taken out at specified time points (4 days for short-term water soaking, and 14 days for long-term water soaking). Then, the water on test specimen surface was wiped, and fixed elongation and tensile tests were performed immediately. For specimens used in fixed elongation test, their fixed elongation maintenance process and elastic recovery process after pad removal were completed under water soaking status.(3)Dry–wet cycle treatments: first of all, the test specimens were soaked in water for 2 days in normal temperature environment. Later, the test specimens were taken out and dried naturally at normal temperature for 2 days. The above steps were repeated 10 times, then, the test specimens were allowed for standing at normal temperature for 7 days before fixed elongation and tensile tests.(4)Ultraviolet radiation treatments: in the tests, the test specimens were transferred into ultraviolet accelerated weathering tester (as shown in Figure 4) for ultraviolet radiation (7 and 14 days for short-term and long-term ultraviolet radiation, respectively). After the completion of radiation, the test specimens were taken out and allowed for standing at normal temperature for 24 h. Finally, fixed elongation and tensile tests were carried out.

Fixed elongation and tensile tests were conducted according to the following methods:

(1) Fixed elongation tests: thefixed elongation test device is shown in Figure 5.

The prepared test specimens were placed into the fixed elongation test device. Then, test specimens were stretched at a rate of 5 mm/min, until their deformation quantity was 60% of the original width. Later, the locating pad was inserted, followed by standing at standard test conditions for 24 h, and the test specimen damage was checked. If the test specimens were not damaged, the pad was removed, the test specimen width was measured, and the elastic recovery rate $R_{e}$ was calculated according to Formula (1).
(1)$$R_{e}=\frac{w_{1}-w_{2}}{w_{1}-w_{0}}\times 100\%$$
where $w_{0}$, $w_{1}$ and $w_{2}$ are the initial width, fixed elongation width and width after elastic recovery of the test specimen, respectively.

(2) Tensile tests: theHS-3001B electronic stretching device (as shown in Figure 6) was adopted to carry out tensile testsat 5 mm/min loading rate.

## 3. Test Results and Analysis

### 3.1. Effects of Temperature

#### 3.1.1. Analysis of Fixed Elongation Property

The typical fixed elongation morphology characteristics of each group of joint sealants at different temperatures are displayed in Figure 7. As observed from Figure 7, all test specimens possessed good fixed elongation property (nocohesion failure or bonding failure) under normal temperature. At the temperature of −10 °C, test specimens in K and Z groups exhibited no obvious failure, while those in Y group showed severe bonding failure. At −20 °C, test specimens in K and Y groups showed bonding failure, while those in Z group still maintained superior fixed elongation bonding performance. The elastic recovery rate of each group is shown in Figure 8. It was illustrated that, at −10 °C, the elastic recovery rate in K and Z groups obviously increased, and Z group had higher elastic recovery rate. At −20 °C, the elastic recovery rate in Z group further increased to 86.7%. Thus, it was obvious that plasticizer addition significantly improved the low temperature flexibility of joint sealants, suchthat they maintained superior fixed elongation property at low temperature.

#### 3.1.2. Analysis of Tensile Property

Figure 9 shows the typical failure morphology of specimen after tensile test. The changes in tensile strength are presented in Figure 10. Clearly, at the temperatures of −20 °C and −10 °C, the tensile strengths of three groups of test specimens were higher than those at normal temperature. Typically, the tensile strength of Z group had the minimum increase amplitude. Compared with test specimens at normal temperature, the tensile strengths of test specimens in K, Y and Z groups at −10 °C increased by 249.4%, 314.2% and 154.4%, respectively. 

Figure 11 illustrates the changes in elongation at break, peak strain, tensile toughness and pre-peak tensile toughness of diverse test specimens at different temperatures. For test specimens in Y group, they rapidly developedfracture failure at the bonding surface after being stretched, so they almost had no deformability (deformation index close to 0). At −20 °C and −10 °C, the elongation at break and peak strain in Z group specimens were greater than those in K group specimens at the same temperature. Test specimens in Y group rapidly developed failure after low temperature stretching, and their tensile toughness sharply decreased at below 0 °C. At this moment, the test specimens almost completely failed at the peak strain, and their pre-peak toughness and tensile toughness were basically the same. As the temperature decreased, test specimens in K and Z groups had increasing tensile toughness, whereas the pre-peak toughness and its percentage kept decreasing. Obviously, plasticizer addition improved the deformation indexes of joint sealants, so that they maintained superior tensile property at low temperature.

#### 3.1.3. Mechanism Analysis

As suggested by the relevant theories in polymer physics [26], the kinematic deformation and conformational change of polymer macromolecular chains under external loads are the chief factors leading to a series of mechanical behaviors with the polymer–cement joint sealants for pavements. In the natural state, the polymer macromolecular chains are often in an irregularly coiled form, at which point the number of molecular conformations and entropy value are the maximum. After application of an external load, the macromolecular chains move and stretch gradually by overcoming the intra-molecular chemical bonding force, the van der Waals force between molecules and the hydrogen-bonding interaction, thereby resulting in diminished number of conformations and decreased entropy value. In macroscopic terms, these are manifested as generation of a certain cohesive strength and displacement deformation. Meanwhile, the loss of adhesion and cohesive failure of joint sealant are the macroscopic representation of the failure and fracture of chemical bonds on the main chain that are oriented by the intermolecular and external forces. After removal of the external load, the molecular chains attempt to restore the coiled state, at which the number of conformations is the largest and the entropy value is the highest, through the internal rotation of single bonds and the segmental motion, thereby endowing the sealant material with a certain elastic recovery property in macroscopic terms. It was figured out that, the environmental temperature significantly affected the mechanical properties of the VAE emulsion–cement composite joint sealants. At different temperatures, the molecular thermal motion degrees of polymer and the excitation energies required for different forms of molecular motion were different. Therefore, there were different degrees of difficulty for the molecular chain segment to overcome the internal rotation barrier. Thus, macroscopically, the fixed elongation and tensile properties of joint sealants exhibited strong temperature sensitivity. At the low temperatures of −10 °C and −20 °C, the polymer molecular thermal motion energy was low, and the chain segment motion was basically at the “freezing” status. As a result, under the action of external load, there were only tiny changes in the side group, chain unit, bond length and bond angle of the macromolecular chain. Macroscopically, it manifested as the decreased flexible deformability of joint sealants and enhanced elastic recovery capacity after removing the external force. Meanwhile, the intermolecular force was enhanced under low temperature environment, and the external load could hardly achieve dissipation equilibrium through stretching motion of molecular chain segment. In this regard, the external load effect was increasingly accumulated inside the joint sealants, which resulted in the rapidly increased cohesion strength of joint sealants macroscopically. Finally, it led to brittle fracture failure or bonding failure at the interface between the joint sealant and the base material bonding surface.

After adding plasticizer, the distance between molecular chains increased, the interaction force was weakened, and the molecular internal rotation barrier decreased. Therefore, the chain segments that could not move under low temperature environment might develop conformational change and stretching motion. Macroscopically, this manifested as a decrease in the cohesion strength of joint sealants added with plasticizer at low temperature, while the flexible deformability increased. After increasing the powder–liquid ratio, the newly added powder particles occupied the free volume between the polymer molecular chains, which increased the steric hindrance and hindered the molecular movement. Consequently, under the action of external load, the motion deformation of the molecular chain segment was obstructed. Macroscopically, this manifested as the increased cohesion strength of joint sealants, reduced deformability, particularly, the low temperature flexibility sharply decreased.

### 3.2. Effects of Water Soaking 

#### 3.2.1. Analysis of Fixed Elongation Property 

The typical fixed elongation morphology characteristics of joint sealants after water soaking treatments are displayed in Figure 12. Under water soaking treatments, no obvious cohesion failure or bonding failure was observed in each group of test specimens, which manifested superior fixed elongation property. The elastic recovery rates of test specimens under water soaking treatments are displayed in Figure 13. It was observed that the elastic recovery rates of each group of test specimens after water soaking slightly decreased. Typically, a longer water soaking time led to a greater decrease amplitude. At the same water soaking time, the elastic recovery rate in Y group had the lowest decrease amplitude, while those in K group and S group had similar decreased amplitude. Under short-term water soaking, the decrease amplitudes in K, Y and S groups were 5.5%, 1.1% and 6.8%, respectively. After wet–dry cycles, the elastic recovery rates of diverse groups of test specimens significantly decreased. Compared with test specimens with no treatment, the elastic recovery rates of test specimens in K, Y and S groups decreased by 11.6%, 18.3% and 14.7%, respectively. 

#### 3.2.2. Analysis of Tensile Property 

The changes in tensile strength are displayed inFigure 14. The tensile strengths of diverse test specimens remarkably decreased after water soaking. A longer time of water soaking resulted in a greater decrease amplitude. Under the same water soaking condition, the tensile strength in S group test specimens had the lowest decrease amplitude, while that in K group test specimens had the greatest decrease amplitude. In short-term water soaking, the tensile strength loss rates of test specimens in K, Y and S groups were 35.1%, 28.3% and 27.8%, respectively. After wet–dry cycle treatment, the tensile strengths of test specimens in all groups significantly increased.

Figure 15 illustratesthe changes in elongation at break, peak strain, tensile toughness and pre-peak tensile toughness of test specimens after water soaking. After water soaking treatment, the elongation at break of test specimens in K group decreased with the extension in water soaking time. By contrast, the elongation at break in Y and S group test specimens increased as the water soaking time extended. Under the same water soaking conditions, the elongation at break of the S group test specimens was greater than that in the Y group test specimen. After water soaking treatment, the peak strains of the three groups of test specimens significantly increased compared with those before water soaking. Afterlong-term water soaking, the peak strains in K, Y and S group test specimens increased by 58.7%, 212.5% and 173.4%, respectively. Typically, the increase in amplitude in K group was the smallest. After water soaking treatment, the tensile toughness of K group test specimens decreased, while the tensile toughness and pre-peak toughness of Y and S group test specimens increased. Thus, it was obvious that, the addition of water repellent and increase in powder–liquid ratio improved the deformation indexes of joint sealants, so that they maintained superior tensile property after water soaking. After wet–dry cycle treatment, the elongation at break slightly decreased, the peak strains partially improved, while the tensile toughness and pre-peak toughness increased.

#### 3.2.3. Mechanism Analysis 

When water entered the interior of the joint sealants, on the one hand, it destroyed the intermolecular hydrogen bonds of the polymer, which resulted in increased distance between molecular chains and weakened intermolecular interaction force. On the other hand, the water dissolving reactionsand hydrolysis reactions further weakened various binding forces inside the joint sealants, which reduced the molecular internal rotation barrier and increased the free volume. With the extension ofwater soaking time and the increase in water infiltration, the above two aspects kept increasing. Therefore, water soaking made it easier for molecular chain segment motion inside the joint sealant, and great displacement deformation was produced under the action of a small external load. Macroscopically, it manifested as the reduced tensile strength of joint sealant after water soaking and significantly increased peak strain, along with accordingly improved pre-peak toughness and its proportion.

After water soaking, the changes in elongation at break and tensile toughness of K group test specimens were slightly different from those in the other two groups. This was because the different mix proportions resulted in the significant change in bonding strengths, failure modes and stress–strain curve morphologies of the joint sealants. For test specimens in K group, the water swelling erosion reduced the bonding strength between the joint sealants and the base materials, as a result, the failure mode of test specimens changed from cohesive failure before water soaking to bonding failure after water soaking. Therefore, the water softening promoted the tensile deformation of joint sealants, but the occurrence of bonding failure prevented the test specimen from being fractured. Additionally, the final deformation degree was small, and the elongation at break decreased. For test specimens in Y group, the increased powder–liquid ratio increased the compactness of joint sealants, thus partially relieving the water swelling erosion. At the same time, the increased cement content further enhanced the bonding strength between joint sealants andcement mortar blocks, which delayed the occurrence of bonding failure. Consequently, the test specimen failure mode after water soaking was the same as that in K group (both were bonding failure). However, the final deformation degree increased due to the accumulation of water softening, leading to the increased elongation at break and tensile toughness of the joint sealants after water soaking. For test specimens in S group, the release of silicyl groups from silicone water repellent effectively improved the hydrophobicity of joint sealants, which suppressed the influence of water swelling erosion on the bonding performance of joint sealants. Thus, the failure modes of the test specimens in S group after water soaking wascohesive failure. Under the action of water softening, the overall deformation degree of test specimens increased compared with that before water soaking, while the elongation at break and tensile toughness of joint sealants increased accordingly. For joint sealant test specimens after dry–wet cycles, a large amount of unhydrated cement particles inside the joint sealant developed secondary hydration reactionswith the infiltrating water. Therefore, the tensile strength of the joint sealant apparently increased, and the deformation flexibility slightly decreased. Notably, after short-term and long-term water soaking, the joint sealants also developed secondary hydration reaction of cement, but its effect was covered by the water softening in the test.

### 3.3. Effects of Ultraviolet Radiation

#### 3.3.1. Analysis of Fixed Elongation Property 

The typical fixed elongation morphology characteristics of joint sealant test specimens in all groups after ultraviolet radiation are presented in Figure 16. It was observed that, after exposure to ultraviolet radiation, no test specimen showed cohesive failure or bonding failure, and test specimens in K, Y and M groups still maintained superior fixed elongation property. The elastic recovery rates of test specimens under ultraviolet radiation are displayed in Figure 17. Under different conditions, the elastic recovery rates of test specimens in K group were the highest, while those in M group were slightly lower than those in K group. After ultraviolet radiation, the elastic recovery rates in all test specimens increased, and a longer ultraviolet radiation time led to a greater increase amplitude. At the same ultraviolet radiation time, the increase amplitude of M group test specimens was the smallest. Of them, at short-term ultraviolet radiation, the increased amplitudes in K, Y and M groups were 5.1%, 7.2% and 2.8%, respectively. Obviously, after the addition of ultraviolet shield agent, the joint sealants still possessed superior fixed elongation property, which effectively resisted the influence of ultraviolet. 

#### 3.3.2. Analysis of Tensile Property 

The changes in tensile strength are shown in Figure 18. As observed from Figure 18, the tensile strengths of test specimens in each group increased after ultraviolet radiation, and a longer ultraviolet radiation time led to a greater increase amplitude. At the same ultraviolet radiation time, the tensile strength in K group experienced the greatest increase, while that in M group had the smallest increase. After short-term ultraviolet radiation, the tensile strengths of test specimens in K, Y and M groups increased by 62.2%, 41.6% and 39.5%, respectively.

Figure 19 illustrates the changes elongation at break, peak strain, tensile toughness and pre-peak tensile toughness of test specimens in each group after ultraviolet radiation.The elongation at break of test specimens in K, Y and M groups decreased with the extension in ultraviolet radiation time, among which, the variation range in Y group was the smallest, while that in K group was the greatest. After long-term ultraviolet radiation, the elongation at break in K, Y and M groups reduced by 14.7%, 5.3% and 11.5%, respectively. The peak strains of test specimens in K, Y and M groups increased with the extension in ultraviolet radiation time, and the test specimens in M group had the smallest increase amplitude at the same radiation time. After ultraviolet radiation, the tensile toughness and pre-peak tensile toughness in K, Y and M groups apparently increased. At the same ultraviolet radiation time, the tensile toughness in Y group had the smallest increase amplitude. Clearly, test specimens in Y group had the most stable tensile propertiesunder the ultraviolet radiation condition, while adding ultraviolet shield agent effectively resisted the impact of ultraviolet radiation on the tensile propertiesof joint sealants.

#### 3.3.3. Mechanism Analysis 

The effects of ultraviolet radiation treatment on the tensile mechanical properties of joint sealants were mainly related to the ultraviolet-induced polymer aging reaction. From the perspective of energy, the short-wave ultraviolet radiation energy was usually greater than the bond energy of most polymer molecules. Therefore, after long-term radiation, a part of the polymer molecules wasin anexcited state and developed photo-oxidation reaction under the synergistic effect of oxygen. Thisreaction gradually developed from the joint sealant surface to the interior, which resulted in cross-linking of polymer molecular chains. Accordingly, the external force required for the polymer molecular chains during the stretching process increased, and the time extended. Consequently, after ultraviolet radiation, the tensile strength, peak strain and various energy consumption indexes of joint sealants apparently increased, but the test specimens rapidly developed failure after reaching the peak strain.

After increasing the powder–liquid ratio, the increase amplitudes of tensile strength and deformation of joint sealants decreased after ultraviolet radiation, suggesting the enhanced ultraviolet radiation aging resistance. This was because that, a greater powder–liquid ratio led to the higher contents of inorganic fillers and cement in the joint sealant, which had better ultraviolet shielding effect. At the same time, more cement hydration products reacted with the polymer active groups, which also further enhanced the joint sealant stability under ultraviolet radiation. Ultraviolet shield agent effectively prevented the penetration of ultraviolet, which reduced the hardening reaction of polymer molecular chains, thus guaranteeing the stable joint sealant properties.

## 4. Durability Analysis

To study the durability of VAE emulsion–cement composite joint sealant, the joint sealant was applied to three Chineseairports. Among them, the airport I is located in Harbin, where high requirements are set forth on the elastic deformability of joint sealant used, especially with regard tothe low temperature flexibility. Airport II is located in Guangzhou, wherehigh requirements are imposed on the heat and water resistances of joint sealant used. Airport III is located in Lhasa, where high requirements are imposed on the aging resistance of the joint sealant used. Under actual operating conditions, the performance of joint sealant wasaffected in a comprehensive way by complex factors, including ultraviolet, water and temperature. From the above test results, it is clear that plasticizer can enhance the flexible deformability of the VAE emulsion–cement composite joint sealants, the addition of water repellent partially improves the water resistance of joint sealants, and the addition of ultraviolet shield agent improves the ultraviolet radiation aging resistance of joint sealants. Based on the above laboratory test results, the mix proportion of H (shown in Table 6) incorporated with plasticizer, water repellent and ultraviolet shield agent is used for field tests. Figure 20 shows main steps of construction of the VAE emulsion–cement composite joint sealant.

In Figure 21, the typical morphologies of the VAE emulsion–cement composite joint sealants are displayed two years after casting. No cohesion failure or bonding failure was observed in the VAE emulsion–cement composite joint sealant, which manifested superior durability. Figure 22 shows the typical morphologies of traditional joint sealant after two years. It can be seen that there are cohesive failure and adhesive failure. Thus, the durability of VAE emulsion–cement composite joint sealant is better than that of traditional joint sealants in service.

The stretching adhesion and elastic deformation properties of the VAE emulsion–cement composite joint sealant conform to the specification and application requirements. Meanwhile, owing to the good chemical stability of inorganic components, the joint sealant has certain performance advantages under some special service conditions (e.g., strong ultraviolet irradiation, etc.). Additionally, the polymer fraction of the composite joint sealants is dominated by water-based polymers. Compared to the oily organic polymer sealants, it causes smaller harmto humans and pollution to the environment during production, construction and later use, which comply better with the green development concept advocated.Thus, the VAE emulsion–cement composite joint sealant is a novel type of joint sealant with outstanding potential.Under actual operating conditions, the performance of joint sealant is affected by complex factors, such as the pavement load, the action frequency and the aviation fuel, etc., in addition to temperature, water, ultraviolet. Hence, the durability of joint sealant was further studied in this paper mainly through field tests. More in-depth research is needed to predict the specific service life of the joint sealant accurately.

## 5. Conclusions 

This paper mainly investigatesthe influence rules of temperatures, water soaking and ultraviolet radiation on the fixed elongation and tensile properties of the VAE emulsion–cement composite joint sealants. In addition, the mechanisms of action of diverse environmental factors wereanalyzed. The main conclusions are drawn below.

(1)At different temperatures, the VAE emulsion–cement composite joint sealants exhibit obvious “low temperature hardening” effect. Compared with normal temperature, the elastic recovery rates, tensile strengths and tensile toughness of joint sealants increase at low temperatures (−10 °C and −20 °C), while various tensile deformation indexes apparently decrease. After adding plasticizer, the cohesive strength of joint sealants at low temperature decreases, while the flexible deformability enhances.(2)After water soaking treatments, the VAE emulsion–cement composite joint sealants possess excellent fixed elongation property, but the elastic recovery rates slightly decreases. After water soaking, the joint sealants become softened under the action of water softening, along with decreased tensile strength and increased peak strain. The addition of water repellent partially improves the water resistance of joint sealants.(3)After ultraviolet radiation, the elastic recovery rates of the VAE emulsion–cement composite joint sealants slightly increase, and they still have good fixed elongation property. Moreover, ultraviolet radiation leads to aging of joint sealants, which remarkably increases their tensile strengths, peak strains and tensile energy consumption indexes. Additionally, the addition of ultraviolet shield agent improves the ultraviolet radiation aging resistance of joint sealants.(4)The increase in powder–liquid ratio reduces the flexibility of polymer–cementcomposite joint sealants, but enhances their water resistance and ultraviolet radiation aging resistance.

## Figures and Tables

**Figure 1 materials-13-03233-f001:**
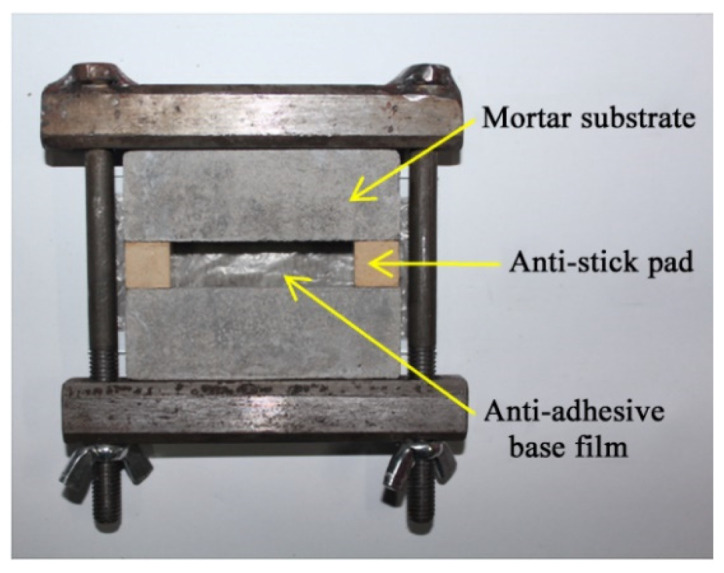
Mold.

**Figure 2 materials-13-03233-f002:**
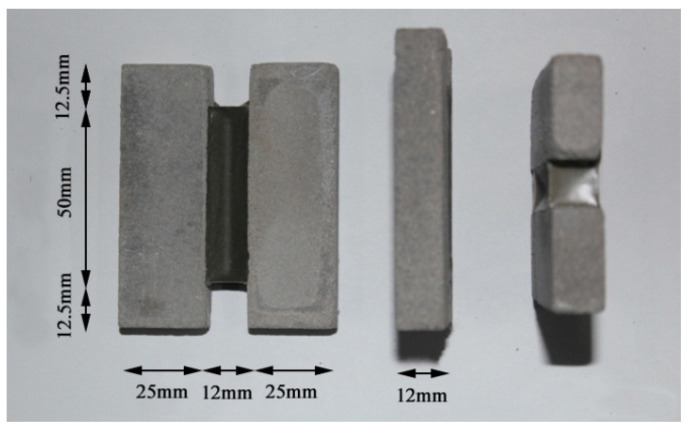
Specimen.

**Figure 3 materials-13-03233-f003:**
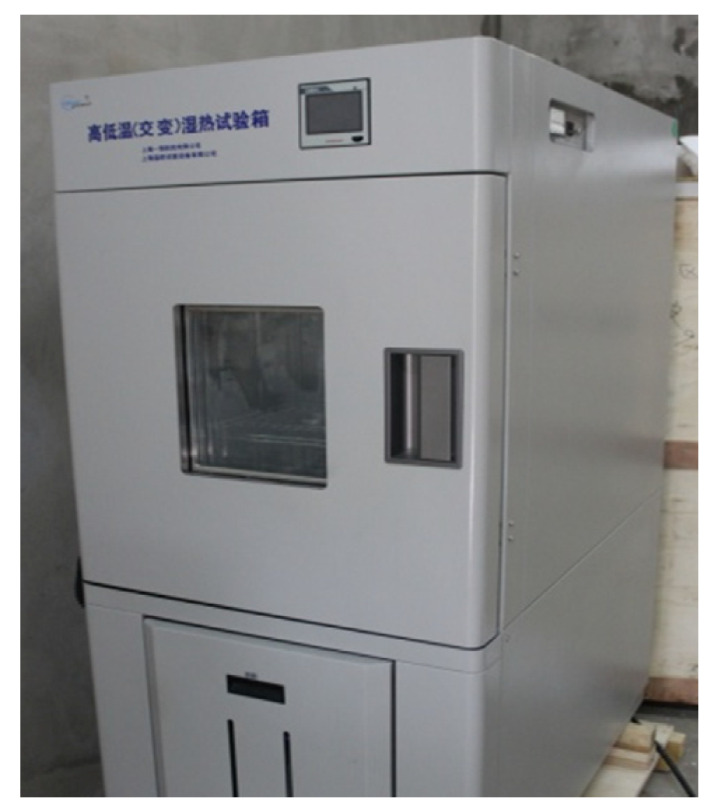
High–low temperature cycle test chamber.

**Figure 4 materials-13-03233-f004:**
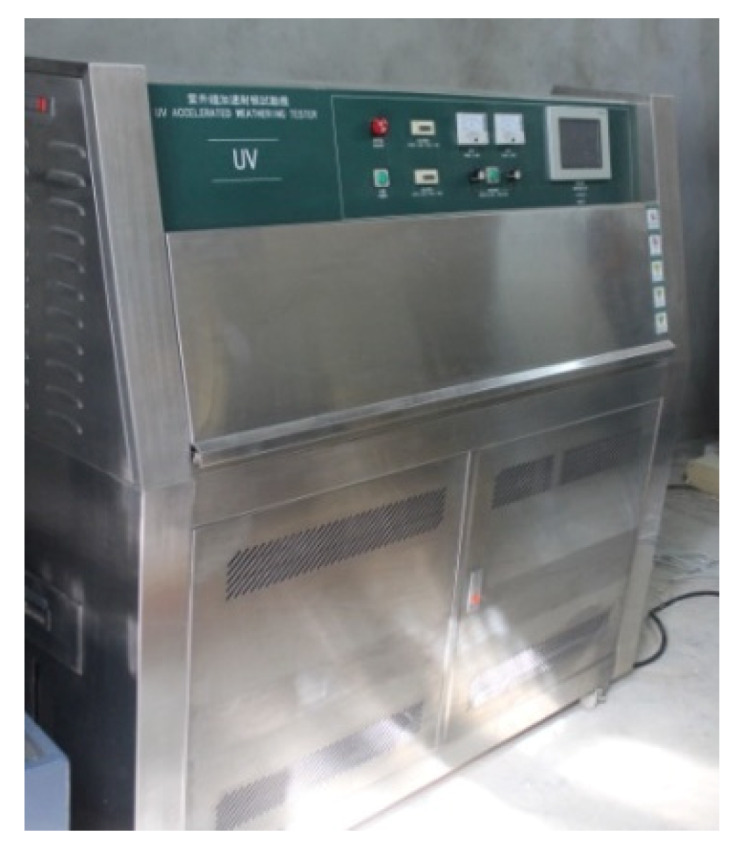
Ultraviolet accelerated weathering tester.

**Figure 5 materials-13-03233-f005:**
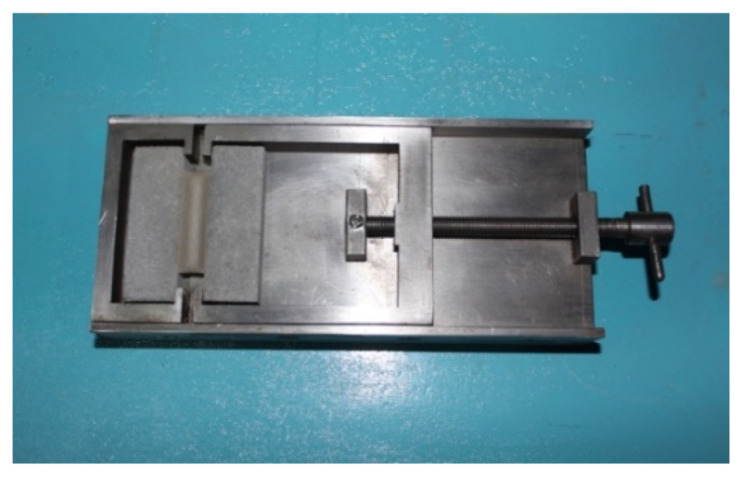
Fixed elongation test device.

**Figure 6 materials-13-03233-f006:**
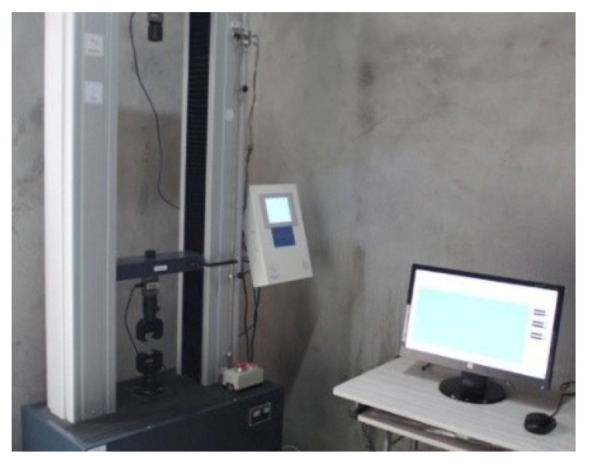
Tensile test device.

**Figure 7 materials-13-03233-f007:**
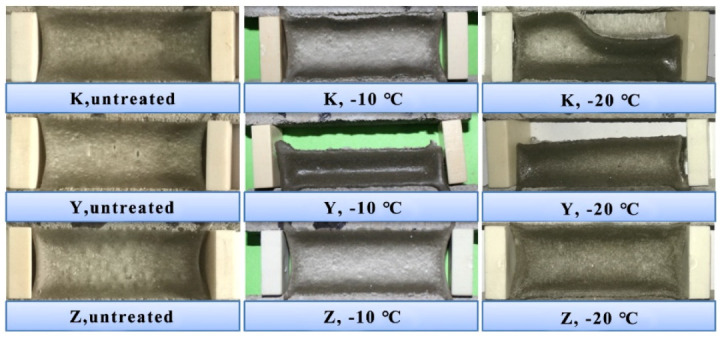
Fixed elongation morphology characteristics oftest specimensat different temperatures.

**Figure 8 materials-13-03233-f008:**
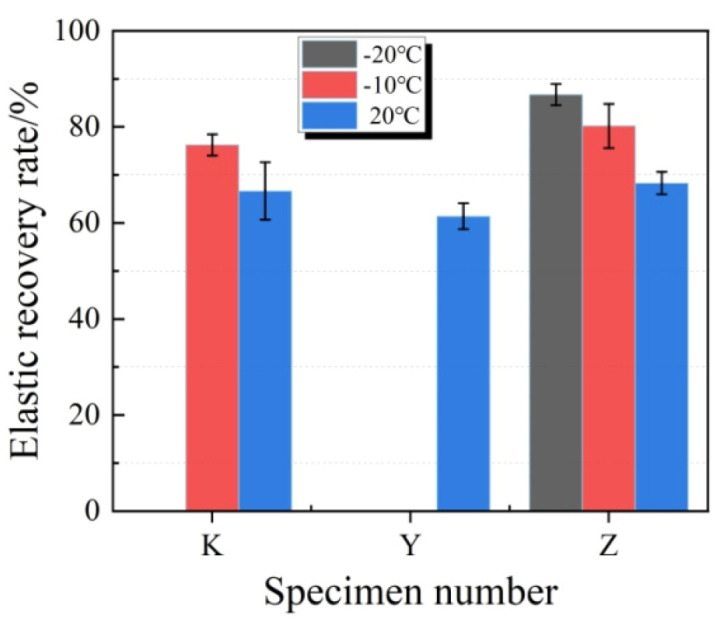
Effects of different temperatures on the elastic recovery rate of test specimens.

**Figure 9 materials-13-03233-f009:**
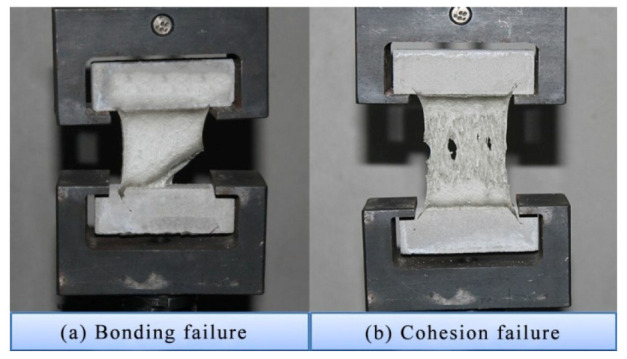
Typical failure morphology of specimen after tensile test. (**a**) Bonding failure; (**b**) Cohesion failure.

**Figure 10 materials-13-03233-f010:**
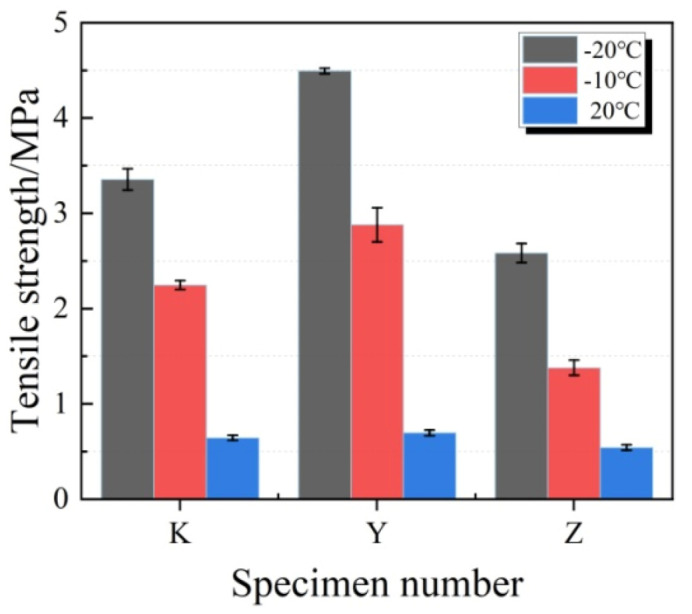
Tensile strength of test specimens at different temperatures.

**Figure 11 materials-13-03233-f011:**
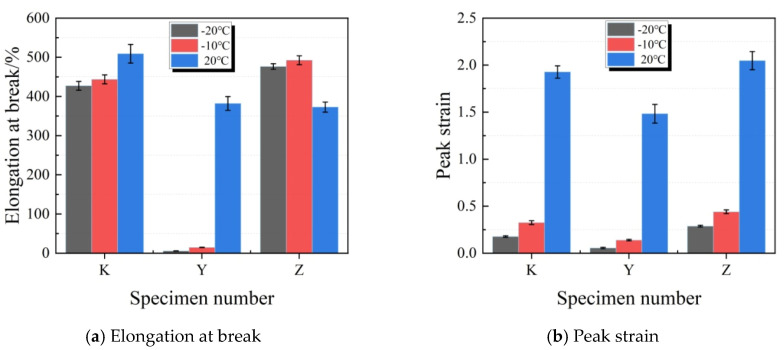
Effects of different temperatures on the tensile property indexes of test specimens. (**a**) Elongation at break; (**b**) Peak strain; (**c**) Tensile toughness; (**d**) Tensile pre-peak toughness.

**Figure 12 materials-13-03233-f012:**
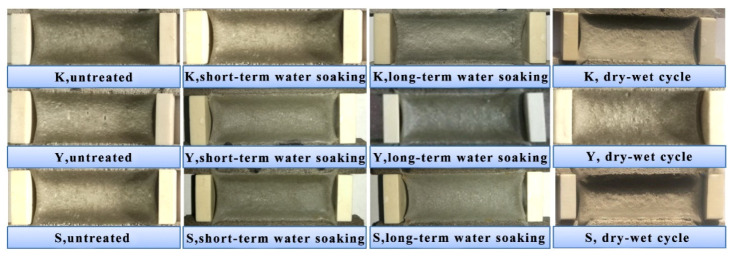
Fixed elongation morphology characteristics oftest specimensunder water soaking treatments.

**Figure 13 materials-13-03233-f013:**
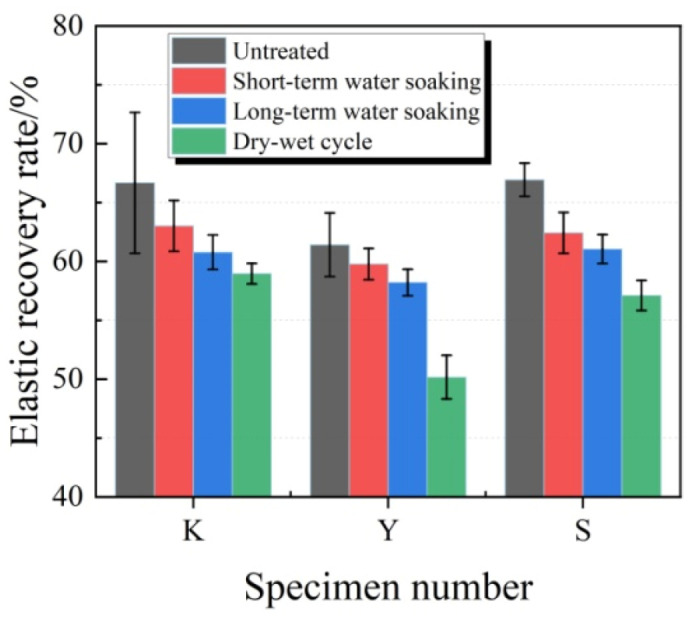
Elastic recovery rates of test specimens under water soaking treatments.

**Figure 14 materials-13-03233-f014:**
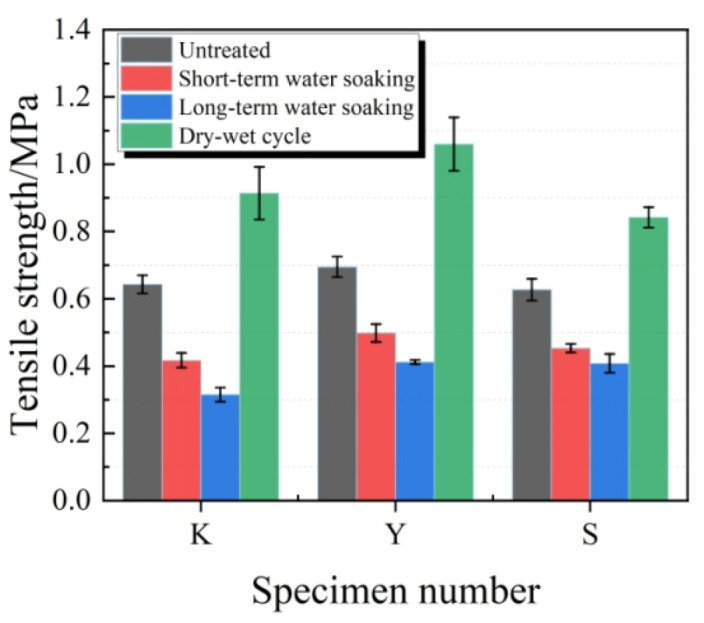
Tensile strength of test specimens under water soaking treatments.

**Figure 15 materials-13-03233-f015:**
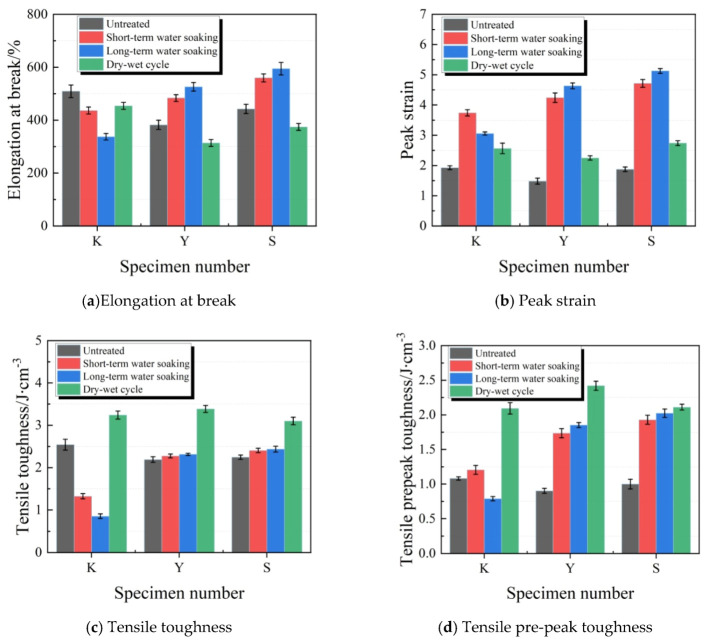
Tensile property indexes of test specimens under water soaking treatments. (**a**) Elongation at break; (**b**) Peak strain; (**c**) Tensile toughness; (**d**) Tensile pre-peak toughness.

**Figure 16 materials-13-03233-f016:**
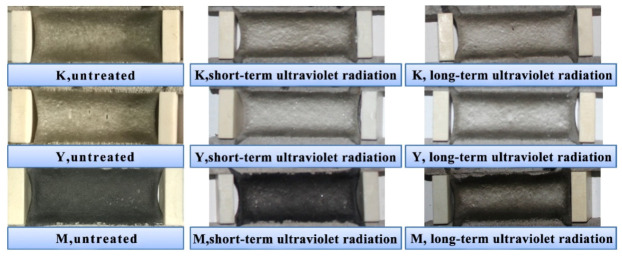
Fixed elongation morphology characteristics of joint sealants under ultraviolet radiation.

**Figure 17 materials-13-03233-f017:**
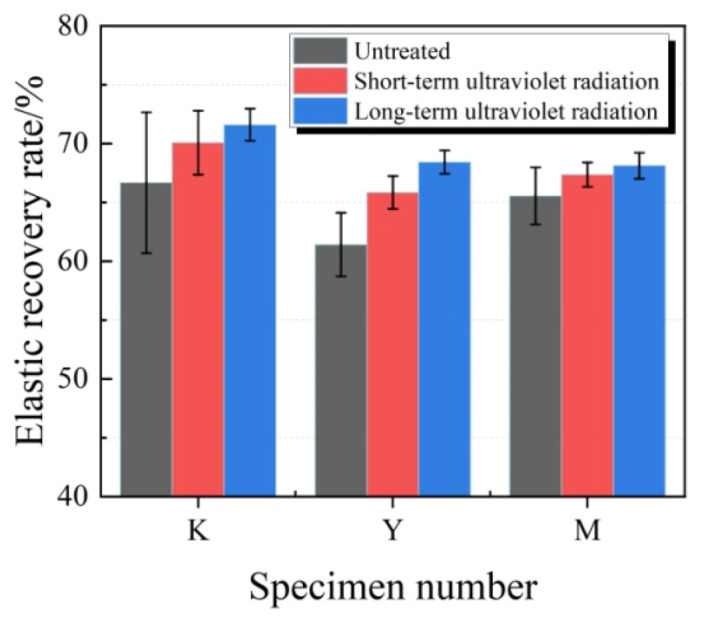
Elastic recovery rates of test specimens under ultraviolet radiation.

**Figure 18 materials-13-03233-f018:**
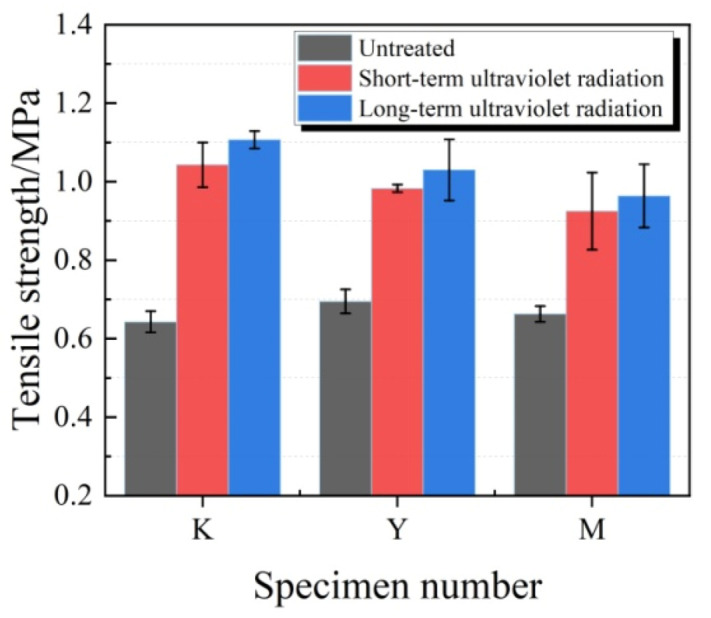
Tensile strength of test specimens under ultraviolet radiation.

**Figure 19 materials-13-03233-f019:**
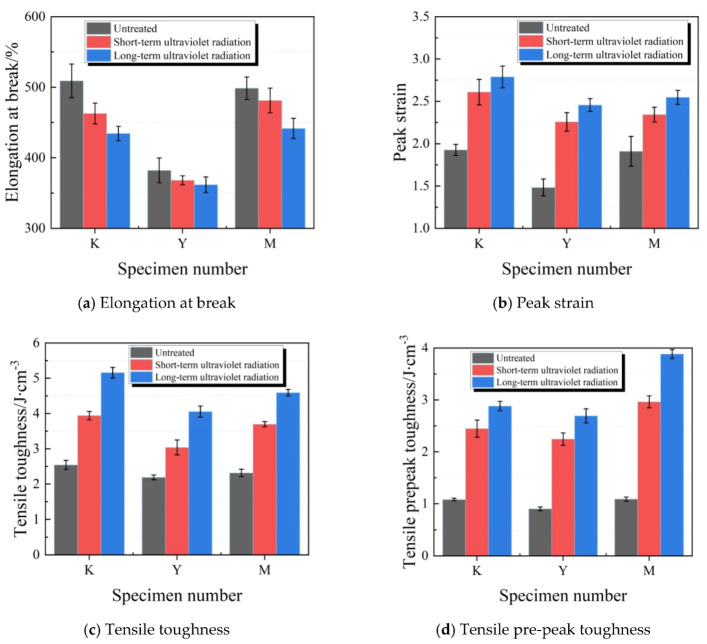
Tensile energy consumption indexes of test specimens under ultraviolet radiation. (**a**) Elongation at break; (**b**) Peak strain; (**c**) Tensile toughness; (**d**) Tensile pre-peak toughness.

**Figure 20 materials-13-03233-f020:**
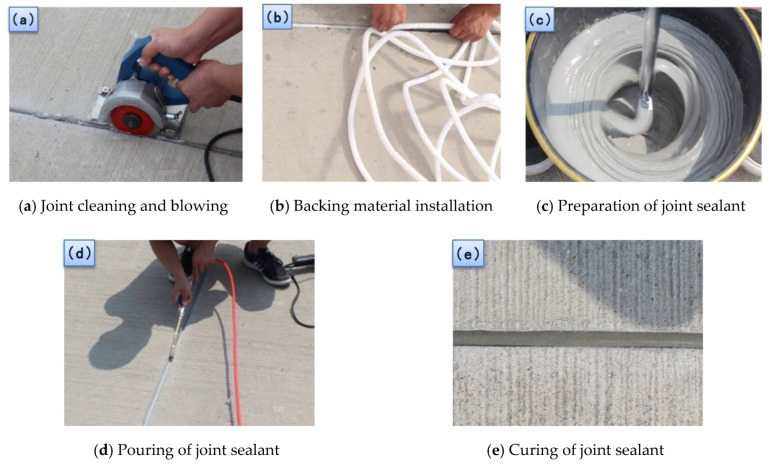
Construction of the VAE emulsion–cement composite joint sealant. (**a**) Joint cleaning and blowing; (**b**) Backing material installation; (**c**) Preparation of joint sealant; (**d**) Pouring of joint sealant; (**e**) Curing of joint sealant.

**Figure 21 materials-13-03233-f021:**
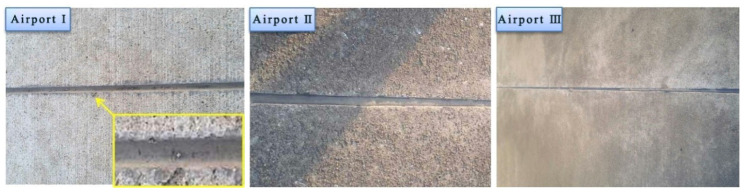
Typical morphologies of the VAE emulsion–cement composite joint sealant after two years.

**Figure 22 materials-13-03233-f022:**
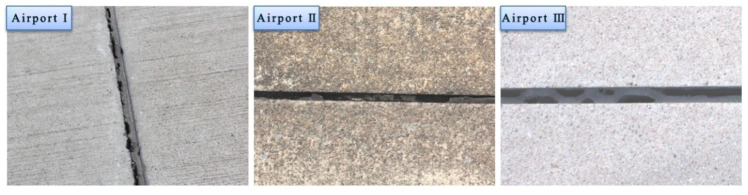
Typical morphologies of traditional joint sealant after two years.

**Table 1 materials-13-03233-t001:** Technical indexes of VAE emulsion.

Appearance	Solid Content/%	Particle Size/μm	Glass-Transition Temperature (T_g_)/°C	Minimum Film Formation Temperature/°C	Brookfield Viscosity/mPa·s
Milk white liquid	54–57	1.5	−6	0	1500–5000

**Table 2 materials-13-03233-t002:** Technical indexes of silicone defoamer.

Appearance	Ionicity	Dispersion	Specific Gravity (25 °C)
Grey liquid	nonionic	insoluble in water	1.02

**Table 3 materials-13-03233-t003:** Technical indexes of dispersing agent.

Appearance	pH	Brookfield Viscosity (25 °C)/mPa·s	Specific Gravity (25 °C)
Light yellow liquid	7.5	450	1.29

**Table 4 materials-13-03233-t004:** Technical indexes of silicone coupling agent.

Appearance	Content/%	Density/g·cm^3^	Boiling Point/°C
Colorless Transparent Liquid	97	0.942	217

**Table 5 materials-13-03233-t005:** Technical indexes of water repellent.

Appearance	Active Ingredient	Apparent Density	pH Value	Residual Moisture
White powder	Siloxane	650–800 g/L	9.0–11.0 (10% aqueous solution)	≤2.0%

**Table 6 materials-13-03233-t006:** Composite mix proportions.

Mix Proportion No.	Powder–Liquid Ratio	Cement ratio	VAE Emulsion/g	Quartz Powder/g	Talc Powder/g	P∙O 42.5 Cement/g	SN-5040 Dispersing Agent/g	SN-345 Defoamer/g	DN-12 Coalescing Agent/g	Coupling Agent/g	Plasticizer/g	Water Repellent/g	Ultraviolet Shield Agent/g
K	0.40	40%	100	14.4	9.6	16	0.98	0.70	6	0.40	/	/	/
Y	0.55	40%	100	19.8	13.2	22	0.98	0.70	6	0.40	/	/	/
Z	0.40	40%	100	14.4	9.6	16	0.98	0.70	6	0.40	3	/	/
S	0.40	40%	100	14.4	9.6	16	0.98	0.70	6	0.40	/	0.70	/
M	0.40	40%	100	14.4	9.6	16	0.98	0.70	6	0.40	/	/	4.8
H	0.40	40%	100	14.4	9.6	16	0.98	0.70	6	0.40	3	0.70	4.8

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
