# Peer review of "Effects of External Environments on the Fixed Elongation and Tensile Properties of the VAE Emulsion–Cement Composite Joint Sealant"

_materials, 2020, doi:10.3390/ma13143233_

Round 1
Reviewer 1 Report
The manuscript describes a complete experimental study of the durability of new formulations of VAE emulsion-cement composite joints sealants. The tests of water soaking, temperature, and UV radiation were performed in the short and long term. Six different mix proportions with the addition of plasticizer, water repellent, and ultraviolet shield agent were prepared and tested. The paper is written in good English and with a clear structure. The topic of the research is meaningful for the practical application and has appropriate scientific soundness for a research paper. I recommend publishing the paper after a minor review including answering the following questions:
- "Figure 21 shows the typical morphologies of traditional joint sealant after two years. It can be seen that there are a cohesive failure and adhesive failure. Thus, the durability of VAE emulsion-cement composite joint sealant is better than that of traditional joint sealants in service." What is this statement based on? Are there any quantitative data indicating the advantages of the VAE emulsion than the traditional joint sealants?
- 2.4. Test methods - is the described testing method related to any standard? If yes, please cite it. If not, please explain why this procedure has been chosen.
- 3. Test results and analysis - how many samples of each mix were tested in each condition? Are the presented results mean values? Please explain.
Author Response
List of Responses
Dear Editors and Reviewers,
We deeply appreciate the time and effort you have spent in reviewing our manuscript entitled “Effects of External Environments on the Fixed Elongation and Tensile Properties of the VAE Emulsion-Cement Composite Joint Sealant” (Manuscript ID: materials-856806). Your comments are all valuable and very helpful for revising and improving our paper. We have studied the comments carefully, and have made revisions point-by-point. All changes made to the previous manuscript are marked in the revised manuscript. The main corrections in the paper and the responds to the reviewers’ comments are as follows.
Responses to the Reviewer #1 comments:
Comment 1:"Figure 21 shows the typical morphologies of traditional joint sealant after two years. It can be seen that there are a cohesive failure and adhesive failure. Thus, the durability of VAE emulsion-cement composite joint sealant is better than that of traditional joint sealants in service." What is this statement based on? Are there any quantitative data indicating the advantages of the VAE emulsion than the traditional joint sealants?
Response: Many thanks for the reviewer’s suggestions. In this manuscript, we analyzed the durability of the VAE emulsion-cement composite sealant and traditional joint sealants based on their morphological changes two years later. According to the field test, we can clearly see failure of the traditional joint sealant, while no apparent change is observed with the VAE emulsion–cement composite sealant, revealing contrasting test results. Given the difficulty of comparing the specific mechanical properties between the traditional joint sealant and the VAE emulsion–cement composite sealant following two years of exposure to the actual environment, we did not conduct quantitative analysis during the experimentation.
Comment 2: Test methods - is the described testing method related to any standard? If yes, please cite it. If not, please explain why this procedure has been chosen.
Response: We thank the reviewer for addressing this issue. We have cited the relevant standards in Section 2.4.
Comment 3: 3. Test results and analysis - how many samples of each mix were tested in each condition? Are the presented results mean values? Please explain.
Response: Thanks for the reviewer’s suggestion. We are very sorry for our failure to give a clear statement of the number of test specimens of each mix which were tested in each condition. Actually, we have conducted repeated tests using three specimens under each condition at each mix, and the test results are the averages of the three repeated test results. We have made a supplementary explanation in the manuscript, and the details are given in Section 2.4.
Once again, we appreciate for Editors/Reviewers’ warm work earnestly, and hope that the revision will meet with approval.
Thank you and best regards.
Sincerely yours,
Chuanxin Lou
Jul 10, 2020
Reviewer 2 Report
The work is written in a very high level in my opinion.
The introduction is sufficient and adequate to the issues considered.
In item 2.1, the material parameters of several materials of the tested composite blends are given in the text, while the others are presented in tabular form (Table 1). You can standardize the form of information about the characteristics of the ingredients used to increase the readability of the information provided.
Table 4 gives the proportions of ingredients used in the tested mixtures. Due to the readability of the work in question, it is worth determining whether the proportions presented refer to the weight or volume of the ingredients used.
It is worth mentioning the number of tests carried out during the tensile tests (similar to row 170 for elongation tests).
The chapter on research and analysis results is clearly legible, the results are presented precisely and with great care. I have no comments for this part of the work.
The final conclusions have been formulated correctly, in relation to the research carried out earlier and their results.
In the conclusions, it is possible to assess which of the tested proportions of the mixture in the authors' opinion are best suited for practical application depending on the prevailing environmental factors (low air temperature, high rainfall rate or strong sunlight during the year).
I request publication of this article after minor corrections.
Author Response
List of Responses
Dear Editors and Reviewers,
We deeply appreciate the time and effort you have spent in reviewing our manuscript entitled “Effects of External Environments on the Fixed Elongation and Tensile Properties of the VAE Emulsion-Cement Composite Joint Sealant” (Manuscript ID: materials-856806). Your comments are all valuable and very helpful for revising and improving our paper. We have studied the comments carefully, and have made revisions point-by-point. All changes made to the previous manuscript are marked in the revised manuscript. The main corrections in the paper and the responds to the reviewers’ comments are as follows.
Responses to the Reviewer#2’s comments:
Comment 1: In item 2.1, the material parameters of several materials of the tested composite blends are given in the text, while the others are presented in tabular form (Table 1). You can standardize the form of information about the characteristics of the ingredients used to increase the readability of the information provided.
Response: Many thanks for the reviewer’s suggestions. As reviewer suggested, We have presented the parameters of test raw materials in the form of table in a uniform manner in Section 2.1. Your comment is greatly valuable in improving the quality of this manuscript.
Comment 2: Table 6 gives the proportions of ingredients used in the tested mixtures. Due to the readability of the work in question, it is worth determining whether the proportions presented refer to the weight or volume of the ingredients used.
Response: Thanks for the reviewer’s suggestion. We apologize for our failure to clarify the unit of material ratio. We have modified the description in Table 6.
Comment 3: It is worth mentioning the number of tests carried out during the tensile tests (similar to row 170 for elongation tests).
Response: Thanks for the reviewer’s suggestion. We are very sorry for our failure to give a clear statement of the number of tests carried out during the tensile tests. Actually, during the tensile tests, we have conducted three repeated tests in each condition at each ratio, and the test results are the averages of the three repeated test results. We have made supplementary explanations in the manuscript, and the details are given in Section 2.4.
Once again, we appreciate for Editors/Reviewers’ warm work earnestly, and hope that the revision will meet with approval.
Thank you and best regards.
Sincerely yours,
Chuanxin Lou
Jul 10, 2020
Reviewer 3 Report
The draft can be accepted in its actual form
Author Response
List of Responses
Dear Editors and Reviewers,
We deeply appreciate the time and effort you have spent in reviewing our manuscript entitled “Effects of External Environments on the Fixed Elongation and Tensile Properties of the VAE Emulsion-Cement Composite Joint Sealant” (Manuscript ID: materials-856806). We appreciate for Editors/Reviewers’ warm work earnestly.
Thank you and best regards.
Sincerely yours,
Chuanxin Lou
Jul 10, 2020
Reviewer 4 Report
The paper presents the results of laboratory tests analysing the influence of different environmental factors on the mechanical properties of the VAE joint sealant.
The results presented should be considered only as preliminary tests which allow to recognize the issue and indicate a direction of future, more detailed, research.
The subject of analysis is interesting but shows generally known trends, which are quite obvious in qualitative terms. Therefore, the results presented should show quantitative analyses of the on the influence of the selected factor on the tested material. In order for the test results to be reliable in the quantitative terms and worth publishing, the research series should allow for statistical analysis for each considered case. Only such preparation of the test results is correct and allows reliable conclusions to be made. The authors tested only one sample of each modified joint sealant, which does not completely validate the obtained result. Tests of individual samples in the series do not show the stability of the results obtained and it is not known whether the result will be the same or similar on subsequent test. Therefore, for all material testing, it is extremely important to provide the mean value of the parameter obtained and its standard deviation. This is not shown here.
Some detailed suggestions and remarks:
- Table 1 ÷3: where the technical information comes from?
- Point 2.2: no information was given about how many samples were tested in a given mixing ratio when determining a particular feature; after the presentation of the results, it can be concluded that only one sample was tested - which is incorrect (general information above).
- Point 2.4: what the adopted test methods were based on? What was the basis for assuming the time of action of the appropriate factor (low temperature, water soaking)?
- Line 138: adopting a low temperature duration of only 24 hours does not completely reflect the reality – see note above.
- point 3.1.1: what does – according to the authors – the statement mean: “good fixed elongation property” ?
- Point 3.1.2: selected samples should be shown after destruction - this will also illustrate the behaviour of the material.
- Sometimes the differences between the tested series are so small (results of fixed elongation) that they can be within the error limit - in the absence of a mean value and standard deviation, this cannot be determined.
- Mechanism analysis: what was the basis of all analysis described? Maybe – verification of changes (on a microscopic scale) in the material structure (before and after testing) or analytical approach to the issue?
- Point 4: How to relate the presented cases of real joint sealant to the laboratory test results obtained? The relationship between laboratory test results and the durability analysis of the practical cases (point 4) must be shown. What is the possibility of predicting the durability of such joint sealant in the conditions of their actual work?
Author Response
List of Responses
Dear Editors and Reviewers,
We deeply appreciate the time and effort you have spent in reviewing our manuscript entitled “Effects of External Environments on the Fixed Elongation and Tensile Properties of the VAE Emulsion-Cement Composite Joint Sealant” (Manuscript ID: materials-856806). Your comments are all valuable and very helpful for revising and improving our paper. We have studied the comments carefully, and have made revisions point-by-point. All changes made to the previous manuscript are marked in the revised manuscript. The main corrections in the paper and the responds to the reviewers’ comments are as follows.
Responses to the Reviewer#4’s comments:
Comment 1:Table 1 -3: where the technical information comes from?
Response: Many thanks for the reviewer’s suggestions. We deeply apologize for our failure to clarify the source of technical information for the test raw materials. Actually, the material technical information come from the manufacturers' product manuals. In the manuscript, the manufacturer and specific model of each material have been described, which are detailed in Point 2.1. Thanks again for your comment, which is greatly valuable in improving the quality of this manuscript.
Comment 2: Point 2.2: no information was given about how many samples were tested in a given mixing ratio when determining a particular feature; after the presentation of the results, it can be concluded that only one sample was tested - which is incorrect (general information above).
Response: Many thanks for the reviewer’s suggestions. We are extremely sorry for our failure to give a clear statement of the number of test specimens which were tested in a given mixing ratio when determining a particular feature. Actually, in the fixed elongation and tensile tests, we have carried out three repeated tests for each specimen in each condition at each ratio, and the test results are the averages of the three repeated test results. We have made supplementary explanations in the manuscript, and the details are given in Point 2.4.
Comment 3: What the adopted test methods were based on? What was the basis for assuming the time of action of the appropriate factor (low temperature, water soaking)?
Response: Many thanks for the reviewer’s suggestions. Our test methods are formulated by consulting national standards. We have given a citation description in Point 2.4.
Comment 4: Line 138: adopting a low temperature duration of only 24 hours does not completely reflect the reality – see note above.
Response: Many thanks for the reviewer’s suggestions. We carried out the low temperature treatment test on the specimens in accordance with the national standards. Although such test procedure allows reflection of the low temperature treatment effect to some extent, it indeed does not completely reflect the reality. Thus, we subjected the specimens separately to the laboratory tests and field tests.
Comment 5: Point 3.1.1: what does – according to the authors – the statement mean: “good fixed elongation property” ?
Response: Many thanks for the reviewer’s suggestions. We deeply apologize for our unclear presentation. "Good fixed elongation property" means that the specimen does not undergo cohesion failure or bonding failure after the tensile test, and the details are given in Point 3.1.1.
Comment 6: Point 3.1.2: selected samples should be shown after destruction - this will also illustrate the behaviour of the material.
Response: Many thanks for the reviewer’s suggestions. In Point 3.1.2, we have added the typical failure morphology of specimen after the tensile test.
Comment 7: Sometimes the differences between the tested series are so small (results of fixed elongation) that they can be within the error limit - in the absence of a mean value and standard deviation, this cannot be determined.
Response: Many thanks for the reviewer’s suggestions. Actually, the test results for each group are the averages of the results of three repeated tests. The average values can reflect the general characteristics of specimens in the same groups, which have a certain persuasive power. We have made a supplementary explanation in Point 2.4.
Comment 8: Mechanism analysis: what was the basis of all analysis described? Maybe – verification of changes (on a microscopic scale) in the material structure (before and after testing) or analytical approach to the issue?
Response: Many thanks for the reviewer’s suggestions. We are very sorry for our failure to clarify the basis for mechanism analysis. Actually, we have carried out the mechanism analysis with reference to the relevant theories in polymer physics, and the details are given in Point 3.1.3. Thanks again for your comment, which is greatly valuable in improving the quality of this manuscript.
Comment 9: How to relate the presented cases of real joint sealant to the laboratory test results obtained? The relationship between laboratory test results and the durability analysis of the practical cases (point 4) must be shown. What is the possibility of predicting the durability of such joint sealant in the conditions of their actual work?
Response: Many thanks for the reviewer’s suggestions. In the actual operating conditions, the performance of joint sealant is affected in a comprehensive way by complex factors, including ultraviolet, water and temperature. From the above test results, it is clear that plasticizer can enhance the flexible deformability of the VAE emulsion-cement composite joint sealants, the addition of water repellent partially improves the water resistance of joint sealants, and the addition of ultraviolet shield agent improves the ultraviolet radiation aging resistance of joint sealants. Based on the above laboratory test results, the specimen incorporated with plasticizer, water repellent and ultraviolet shield agent at a mix proportion of H is used for field tests. In the actual operating conditions, the performance of joint sealant is affected by complex factors, such as the pavement load, the action frequency and the aviation fuel, etc aside from temperature, water, ultraviolet. Hence, the durability of joint sealant was further studied in this paper mainly through field tests. More in-depth research is needed to predict the specific service life of the joint sealant accurately. We have made a supplementary explanation in Point 4. Thanks again for your comment, which is greatly valuable in improving the quality of this manuscript.
Once again, we appreciate for Editors/Reviewers’ warm work earnestly, and hope that the revision will meet with approval.
Thank you and best regards.
Sincerely yours,
Chuanxin Lou
Jul 10, 2020
Round 2
Reviewer 4 Report
I thank the authors for your answers and supplement the article with the necessary information.
Unfortunately, the authors still did not provide the standard deviation (or at least the coefficient of variation) of the obtained average results, which - in my opinion - is necessary here. I find this a drawback to the paper. Supplementing this information will definitely improve the quality of the article.
Author Response
List of Responses
Dear Editors and Reviewers,
We deeply appreciate the time and effort you have spent in reviewing our manuscript entitled “Effects of External Environments on the Fixed Elongation and Tensile Properties of the VAE Emulsion-Cement Composite Joint Sealant” (Manuscript ID: materials-856806). Your comments are very valuable and helpful for revising and improving our paper. We have studied the comments carefully, and have made revisions. All changes made to the previous manuscript are marked in the revised manuscript. The main corrections in the paper and the responds to the reviewers’ comments are as follows.
Responses to the Reviewer#4’s comments:
Comment 1: Unfortunately, the authors still did not provide the standard deviation (or at least the coefficient of variation) of the obtained average results, which - in my opinion - is necessary here. I find this a drawback to the paper. Supplementing this information will definitely improve the quality of the article.
Response: Many thanks for the reviewer’s suggestions. The error bar of the obtained average results has been added in figures of Point 3. Thanks again for your comment, which is greatly valuable in improving the quality of this manuscript.
Once again, we appreciate for Editors/Reviewers’ warm work earnestly, and hope that the revision will meet with approval.
Thank you and best regards.
Sincerely yours,
Chuanxin Lou
Jul 16, 2020
This manuscript is a resubmission of an earlier submission. The following is a list of the peer review reports and author responses from that submission.